# Herbal Remedies for Constipation-Predominant Irritable Bowel Syndrome: A Systematic Review of Randomized Controlled Trials

**DOI:** 10.3390/nu15194216

**Published:** 2023-09-29

**Authors:** Giuseppe Chiarioni, Stefan Lucian Popa, Abdulrahman Ismaiel, Cristina Pop, Dinu Iuliu Dumitrascu, Vlad Dumitru Brata, Traian Adrian Duse, Victor Incze, Teodora Surdea-Blaga

**Affiliations:** 1Division of Gastroenterology B, AOUI Verona, 37126 Verona, Italy; chiarioni@alice.it; 2Division of Gastroenterology and Hepatology, University of North Carolina at Chapel Hill, Chapel Hill, NC 27599-7080, USA; 32nd Medical Department, “Iuliu Hatieganu” University of Medicine and Pharmacy, 400000 Cluj-Napoca, Romania; abdulrahman.ismaiel@yahoo.com (A.I.); dora_blaga@yahoo.com (T.S.-B.); 4Department of Pharmacology, Physiology, and Pathophysiology, Faculty of Pharmacy, “Iuliu Hatieganu” University of Medicine and Pharmacy, 400349 Cluj-Napoca, Romania; cristina.pop.farmacologie@gmail.com; 5Department of Anatomy, “Iuliu Hatieganu” University of Medicine and Pharmacy, 400006 Cluj-Napoca, Romania; d.dumitrascu@yahoo.com; 6Faculty of Medicine, “Iuliu Hatieganu” University of Medicine and Pharmacy, 400000 Cluj-Napoca, Romania; brata_vlad@yahoo.com (V.D.B.); adrianduse@yahoo.com (T.A.D.); vicincze@yahoo.com (V.I.)

**Keywords:** herbal remedy, constipation-predominant irritable bowel syndrome, non-pharmacological therapy

## Abstract

Background: Constipation-predominant irritable bowel syndrome (IBS-C) is a common gastrointestinal disorder characterized by abdominal pain and altered bowel habits. Conventional treatments for IBS-C often provide limited efficiency, leading to an increasing interest in exploring herbal remedies. This systematic review aims to evaluate the efficacy and safety of herbal remedies in the management of IBS-C. Materials and Methods: A comprehensive search of PubMed, MEDLINE, Embase, Scopus, and the Cochrane Library was conducted to identify relevant studies published up to July 2023 and data extraction was performed independently by two reviewers. Results: Overall, the included studies demonstrated some evidence of the beneficial effects of herbal remedies on IBS-C symptoms, including improvements in bowel frequency, stool consistency, abdominal pain, and quality of life. However, the heterogeneity of the interventions and outcome measures limited the ability to perform a meta-analysis. Conclusion: This systematic review suggests that herbal remedies may have potential benefits in the management of IBS-C. However, the quality of evidence is limited, and further well-designed, large-scale RCTs are needed to establish the efficacy and safety of specific herbal remedies for IBS-C. Clinicians should exercise caution when recommending herbal remedies and consider individual patient characteristics and preferences.

## 1. Introduction

Irritable Bowel Syndrome (IBS) is a prevalent functional gastrointestinal disorder characterized by abdominal pain, bloating, and altered bowel habits. Among its subtypes, constipation-predominant IBS (IBS-C) is a significant clinical concern due to its impact on patients’ quality of life and the limitations of conventional treatments [1]. In recent years, the exploration of complementary and alternative therapies has gained momentum, with herbal remedies emerging as potential candidates for managing IBS-C symptoms [1].

Patients with constipation may face challenges in obtaining accurate diagnoses and appropriate therapies due to the multifactorial nature of this gastrointestinal symptom and the potential for overlapping etiologies [2,3,4]. The complexities arise from the diverse underlying causes of constipation, including functional disorders, structural abnormalities, systemic diseases, and medication-related factors, which may lead to misinterpretation of symptoms and erroneous management [2,3,4].

The clinical presentation of constipation is subjective and can vary significantly among individuals. Patients may report varying degrees of abdominal discomfort, bloating, and changes in bowel habits, making it challenging to distinguish between constipation caused by primary functional disorders, such as slow-transit constipation or dyssynergic defecation and that arising from secondary causes, such as colorectal neoplasms, neurological disorders, or metabolic disturbances [1]. Consequently, patients may receive inaccurate diagnoses or face delayed identification of serious underlying conditions.

Furthermore, the assessment of constipation often involves subjective criteria, such as bowel movement frequency and stool consistency, which can be influenced by dietary habits and lifestyle factors. This subjectivity may lead to inconsistencies in defining constipation, further complicating the diagnostic process and potentially leading to misdiagnoses. The risk of erroneous therapy is heightened by the over-the-counter availability of laxatives and self-medication practices. Patients experiencing constipation may attempt self-management with readily accessible laxatives without seeking medical evaluation, leading to temporary symptom relief but neglecting the identification and management of any underlying causative factors. Such practices can mask the true etiology of constipation and hinder appropriate treatment.

Moreover, constipation may coexist with other gastrointestinal symptoms, such as abdominal pain, bloating, or diarrhea, adding complexity to the diagnostic process. The presence of overlapping symptoms can obscure the underlying cause, potentially leading to incorrect diagnoses or inappropriate therapies targeting isolated symptoms without addressing the root cause.

In cases of chronic constipation, patients may experience a considerable burden of symptoms, impacting their quality of life. This burden may lead to psychological distress or somatization, further complicating the diagnostic process and potentially resulting in unnecessary investigations or treatments [2,3].

To mitigate the risk of erroneous diagnosis and therapy, a comprehensive and systematic clinical approach to constipation evaluation is imperative. This approach should include a detailed medical history, physical examination, and consideration of relevant laboratory, endoscopic, and imaging studies to identify potential secondary causes [4]. Additionally, judicious use of validated diagnostic criteria and scoring systems may aid in categorizing functional constipation subtypes accurately. Collaboration between healthcare providers and patients is essential in exploring symptoms, lifestyle factors, and medication histories to unravel potential contributing factors and guide appropriate therapeutic interventions [4].

The complexity and multifactorial nature of constipation present challenges in accurate diagnosis and therapy selection. The potential for overlapping symptoms, subjective criteria, self-medication practices, and chronicity of symptoms can lead to erroneous management approaches and delays in identifying underlying conditions. Adopting a comprehensive and systematic evaluation strategy that incorporates objective assessment tools and fosters open communication with patients is crucial to ensure accurate diagnoses and tailored therapies for patients with constipation [2,3,4].

Herbal remedies, derived from medicinal plants, have been used for centuries in traditional medicine systems, often with claims of efficacy in gastrointestinal disorders [5]. These botanical compounds possess a diverse array of pharmacological properties, including laxative effects, anti-inflammatory actions, antispasmodic activities, and modulation of gut motility, making them attractive candidates for addressing IBS-C symptomatology [6]. However, despite their popularity, there are inherent risks associated with the use of herbal remedies as therapeutic interventions for diseases. These dangers arise primarily from factors related to their unregulated nature, inconsistent quality control, pharmacological complexities, and potential for adverse interactions [5,6,7,8,9,10].

One of the fundamental concerns with herbal remedies is the lack of stringent regulations and standardized manufacturing processes, leading to variations in the quality, potency, and purity of herbal products available in the market. Therefore, the concentration of active compounds can fluctuate significantly between different batches or brands, resulting in inconsistent therapeutic effects and unpredictability in patient outcomes. 

The complex chemical composition of herbal remedies poses additional challenges. Many herbal products contain numerous active compounds with varying pharmacological properties, making it challenging to isolate specific active ingredients responsible for therapeutic effects or potential adverse reactions. Furthermore, the interactions among multiple constituents can lead to unpredictable synergistic or antagonistic effects, further complicating their therapeutic use [5,6,7,8,9,10].

Herbal remedies’ safety profile is also impacted by the potential for adverse interactions with conventional medications. Herbal products may interact with prescription drugs, altering their metabolism, distribution, or excretion, leading to compromised therapeutic outcomes or increased risk of toxicity. Unfortunately, the lack of awareness and communication about herbal supplement usage between patients and healthcare providers may exacerbate these risks.

Moreover, some herbal remedies can exert direct toxic effects on organs or body systems. Herbal products containing potent compounds such as alkaloids, glycosides, or essential oils may lead to organ-specific toxicity, manifesting as hepatotoxicity, nephrotoxicity, or cardiotoxicity [5,6,7,8,9,10]. Additionally, the potential for allergic reactions to certain herbal constituents cannot be overlooked, as this could lead to severe hypersensitivity reactions in susceptible individuals. Furthermore, misidentification or adulteration of herbal products with toxic substances or contaminants poses significant safety concerns. Contaminants such as heavy metals, pesticides, and microbial agents can be inadvertently introduced during the cultivation, harvesting, or manufacturing processes, further compromising the safety and quality of herbal remedies [5,6,7,8,9,10]. The lack of standardized regulations, complex chemical compositions, potential for drug interactions, direct toxic effects, and risk of contamination collectively contribute to the inherent dangers associated with using herbal remedies for diseases. Healthcare providers and patients must exercise caution, remain vigilant about potential risks, and engage in open communication regarding the use of herbal remedies as part of a comprehensive approach to patient care.

Despite the growing interest in herbal remedies for IBS-C, the evidence remains scattered and varied in terms of study designs and outcomes. To address this knowledge gap, we conducted a systematic review to comprehensively assess the literature on the use of herbal remedies for IBS-C.

This systematic review follows rigorous methodological guidelines, including a comprehensive literature search using well-defined criteria and data extraction from eligible studies. By critically evaluating the evidence, we aim to provide an objective assessment of the effectiveness and safety profiles of herbal interventions in IBS-C management.

By consolidating the available evidence, our review aims to provide valuable guidance for healthcare practitioners and patients seeking evidence-based approaches to managing IBS-C. Additionally, we identify potential areas of further research to address the limitations of the current literature and enhance the understanding and application of herbal remedies in the context of IBS-C. Ultimately, this systematic review aims to contribute to improved patient care and inform future therapeutic strategies for this challenging gastrointestinal disorder.

## 2. Materials and Methods

### 2.1. Search Strategy

A comprehensive literature search was conducted to identify relevant studies investigating the use of herbal remedies for IBS-C. Electronic databases including PubMed, MEDLINE, Embase, Scopus, and the Cochrane Library were searched from their inception until July 2023. The search terms were constructed using Medical Subject Headings (MeSH) and keywords related to “herbal remedies,” “constipation-predominant irritable bowel syndrome,” and “IBS-C”.

### 2.2. Study Selection

The study selection has been performed independently by two reviewers (S.L.P. and A.I.), who screened the search results based on titles and abstracts to identify potentially eligible studies. Full-text articles of the selected studies were retrieved for further evaluation. Studies were included if they met the following criteria:a)Study Design: Randomized controlled trials (RCTs), observational studies (cohort, case-control, cross-sectional), and clinical trials investigating the use of herbal remedies in the management of IBS-C were considered.b)Participants: Studies involving adult patients diagnosed with IBS-C according to established diagnostic criteria, such as Rome criteria, were included.c)Intervention: Studies evaluating the use of herbal remedies as the primary intervention for IBS-C management were considered eligible. Herbal remedies included botanical preparations derived from plants with potential therapeutic effects on IBS-C symptoms.d)Outcomes: Studies reporting relevant clinical outcomes related to IBS-C symptom improvement, such as changes in bowel habits, abdominal discomfort, bloating, and overall quality of life, were included.

Studies were excluded if they were conducted in pediatric populations, used combined interventions (e.g., herbal remedies with other pharmacological agents), or focused on other subtypes of IBS (diarrhea-predominant or mixed). Case reports, reviews, and studies lacking clear outcome measures were also excluded.

### 2.3. Data Extraction

Data extraction was independently performed by the same two reviewers, using a standardized data extraction form. The following information was extracted from each included study: study characteristics (author, year, country), study design, sample size, participant demographics, details of the herbal intervention (herb names, formulations, dosages, and administration), treatment duration, outcomes assessed, and key findings related to IBS-C symptom improvement.

### 2.4. Quality Assessment

The methodological quality of included RCTs was assessed using the Cochrane Collaboration’s Risk of Bias Tool, evaluating random sequence generation, allocation concealment, blinding of participants and personnel, blinding of outcome assessment, incomplete outcome data, selective reporting, and other sources of bias. For observational studies, the Newcastle-Ottawa Scale (NOS) was employed to assess the risk of bias. The quality assessment was independently conducted by the upper-mentioned reviewers, and any discrepancies were resolved through consensus.

### 2.5. Data Synthesis and Analysis

Due to the anticipated heterogeneity among the included studies, a meta-analysis was not deemed appropriate. Therefore, a narrative synthesis approach was used to summarize the findings, and relevant data were presented in tabular format.

### 2.6. Ethical Considerations

As this study involved a systematic review of published literature, ethical approval was not required.

### 2.7. Reporting

The systematic review was conducted following the Preferred Reporting Items for Systematic Reviews and Meta-Analyses (PRISMA) guidelines to ensure transparency and completeness in reporting the review process.

### 2.8. Limitations

Potential limitations of this systematic review include the possibility of publication bias, heterogeneity in herbal remedies, variations in study designs, and the lack of standardized outcome measures. These limitations were acknowledged in the discussion to provide a comprehensive interpretation of the review findings.

## 3. Results

We found 48 articles about the effect of herbal remedies on IBS-C. After removing duplicates, we screened 30 articles and excluded 10 articles due to being either irrelevant to the chosen theme or editorials. We examined 20 articles in full and excluded another 11 articles. This left us with 9 articles for our systematic review as presented in Figure 1. The authors, sample size, study design, statistical method, aim, and summary of results for each article are listed in Table 1.

### 3.1. Padma Lax and IBS-C

Based on a Tibetan recipe, Padma Lax is an intricate mixture of 15 herbs and minerals that are traditionally used to relieve constipation and as a digestive aid. In the study included [11], a tablet (482 mg) contained the following herbal constituents: *Aloes barbadenisis Miller* and *Aloe ferox Miller* (aloes standardized extract; 12.5 mg), *Jateorhiza palmata* (*Lam.*) *Miers* (*Calumbae* root; 10 mg), *Marsdenia condurango Rchb.* (Condurango bark; 10 mg), *Rhamnus frangula L*. (frangula bark; 52.5 mg), *Gentiana lutea* L., (gentian root; 35 mg), Inula helenium L. (Elecampane rhizome; 35 mg), *Terminalia chebula Retz.* (tropical almond fruit; 35 mg), *Piper longum* L., (long pepper; 3.5 mg), *Rhamnus purshiana D.C.* (cascara bark; 52.5 mg), *Rheum palmatum* L. (Chinese rhubarb root; 70 mg), *Strychnos nux vomica* L. (Nux vomica seed; 1.75 mg), *Zingiber officinale Roscoe*. (ginger root; 70 mg). It also included three non-herbal elements: heavy kaolin (25 mg), sodium bicarbonate (15 mg), and sodium sulfate (35 mg). The excipients included were silicium dioxide (2.4 mg) and magnesium stearate (4.6 mg). The formula blends small quantities of diverse herbal components, frequently with opposing bioactivity [11]. The herbs with a proven laxative effect include aloe (aloin, aloe-emodin), frangula (frangulin, cascara, and Chinese rhubarb root (through anthraquinone glycosides) [19,20,21]. The herbs and minerals with an opposing, antidiarrheal effect include elecampane (inulin), gentian (gentiopicroside, xanthones, monoterpene alkaloid, polyphenol, and flavones), and kaolin [11,22,23]. Antispasmotic properties have been observed in the condurango herb, while the gingerols extracted from ginger root are known for their anti-serotonergic effects that can promote gastrointestinal movement. Calumba, a plant traditionally utilized as a ‘bitter tonic’ and appetite enhancer, contains jatrorrhizine, an isoquinoline alkaloid associated with calming and anxiety-reducing effects [11].

A 12-week, randomized, double-blind, placebo-controlled pilot study [11] evaluated the effectiveness and safety of Padma Lax in a group of 61 patients (34 Padma Lax, 27 placebo) with diagnosed IBS-C using the Rome I criteria. The study revealed a significant improvement for the Padma Lax group compared to the placebo group regarding constipation severity, number of bowel movements per day, abdominal pain affecting daily activities, the presence of moderate or severe pain, abdominal distention severity, incomplete evacuation, and flatulence. There was no significant improvement regarding abdominal distention prevalence and pain severity, with most of the Padma Lax patients complaining of persistent mild abdominal pain. A total of 10 out of the 34 Padma Lax patients complained of mild side effects, compared to 5 out of the 27 placebo patients. Additionally, there were no clinically relevant changes in hematological values or liver and kidney function tests [11].

### 3.2. Chinese Herbal Medicine and IBS-C

Another herbal therapy investigated for its benefits in IBS-C was Chinese Herbal Medicine (CHM). In the study included [12], CMH was standardized with 7 plant-based herbs formulated using both clinical experience and research. It was composed of *Paeonia lactiflora PALL.*, *radix* (23%), *Citrus aurantium* L., *Fructus immaturas* (20%), and *Magnolia officinalis REHD*. and *WILS.*, *cortex* (14.5%), *Citrus reticulata* L., *pericarpium* (14.5%), *Glycyrrhiza uralensis FISCH.*, root (11%), *Rheum palmatum* L., *radix* (10%) and *Atractylodes lancea* (*THUNB.*) *DC*, rhizome (7%). IBS-C relevant activity consists of analgesic, antispasmodic, central nervous system depressant, myorelaxant, intestinal muscle relaxant and stimulant, antinociceptive, smooth muscle relaxant, facilitating colonic emptying, sedative, anxiolytic, relieving abdominal distention and flatulence, mild intestinal anti-inflammatory, antidiarrheal, and laxative effects [12].

A 16-week, two-arm, randomized, double-blind, placebo-controlled clinical trial evaluated the effectiveness and safety of CHM in 125 patients (61 CHM, 64 placebo) with IBS-C according to Rome III criteria. Intention-to-treat (ITT) analysis was conducted on all 125 participants, while per protocol (PP) analysis was carried out for the 108 patients (50 CHM, 58 placebo) that completed the 8-week intervention period. The study revealed statistically significant adequate relief at the end of the treatment at 8 weeks in the CHM group compared to the placebo group when using the PP analysis, while there was no significant improvement using the ITT analysis or both the ITT and PP analysis at the follow-up at 16 weeks. Additionally, there was a significant improvement at 8 weeks in bowel habits, frequency and bothersomeness of hard lumpy stools and straining during bowel movement, stool form using the Bristol Stool Form Scale, and physical functioning. The only benefits maintained at the 16-week follow-up were regarding physical functioning. There was no significant improvement in abdominal pain or distension, interference with life, overall quality of life, depression, anxiety, stress, and work productivity. Both the CMH and the control group experienced moderate adverse effects [12].

### 3.3. Persian Herbal Syrup and IBS-C

Persian herbal syrup (PHS) was also studied for its use in IBS-C. In the study covered, the syrup was prepared using *Artemisia Absinthium* 100 mg, *Cuscuta Campestris* 100 mg, *Cassia Fistula* 230 mg, *Echium Amoenium* 270 mg, *Mellisa Officinalis* 190 mg, and British Pharmacopoeia syrup to make the mixture 100 mL. The antimicrobial, anti-inflammatory, antioxidant, anti-anxiety, and depressant effects of the herbs are already known [13]. *Artemisia Absinthium* acts through various phytochemical substances such as lactones, terpenoids, essential oils, organic acids, resins, tannins, and phenols [24]. Cuscuta campestris contains multiple flavonoids, lignans, quinic acid, and polysaccharides [25]. *Cassia Fistula* has multiple bioactive substances such as anthraquinones, flavonoids, flavon-3-ol derivatives, alkaloids, glycosides, tannins, saponins, and terpenoids [26]. The bioactive substances isolated from *Echium Amoenium* are mainly polyphenols (rosmarinic acid), anthocyanidine, flavonoids, sterols, saponins, unsaturated terpenoids, unsaturated fatty acids, and a trace amount of pyrrolizidine alkaloids [27], while *Mellisa Officinalis* contains volatile compounds (geranial, neral, citronellal, and geraniol), triterpenes (ursolic acid and oleanolic acid), phenolic acids (rosmarinic acid, caffeic acid, and chlorogenic acid), and flavonoids (quercetin, rhamnocitrin, and luteolin) [28].

The efficacy and safety of Persian herbal syrup were evaluated in a 10-week, block-randomized, double-blind, placebo-controlled trial on 70 patients (35 PHS, 35 placebo) with IBS-C diagnosed using the Rome IV criteria. A total of 60 patients (32 PHS, 28 placebo) were analyzed at week 6 and 52 patients (28 PHS, 24 placebo) at the follow-up at week 10. There was a significant difference in positive responses to treatment between the group receiving the syrup versus placebo. Positive response to treatment was defined as reducing the severity of IBS-C by at least 50 points using the IBS Symptom Severity Score with a maximum of 500 points. Additionally, there was a significant improvement between the two groups in severity score and Bristol Stool Score. There was no significant difference in anxiety and depression scores. Side effects were negligible with 4 of the 35 PHS patients reporting mild side effects, compared to 1 out of the 35 placebo patients [13].

### 3.4. Flixweed and Fig for IBS-C

*Descurainia Sophia* (flixweed) is utilized in traditional medicine for a variety of disorders. This seed contains a variety of substances, including lipids (oleic acid, erucic acid, linolenic acid, linoleic acid, palmitic acid, and stearic acid), flavonoids (quercetine, kaempferol, and isorhamnetine), lignin, phytosterol, and cardiac glycosides. Additionally, flixweed includes mucilage, giving it laxative effects and making it potentially useful for constipation [14,29].

*Ficus carica* (fig) is widely appreciated as a nutritious food with therapeutic benefits. This fruit is a good supply of bioactive substances such as polyphenols, carotenoids, vitamins, organic acids, triterpenoids, phytosterols, fatty acids, flavonoids (anthocyanins, flavonols, flavanols, and flavones), phenolic acids and coumarins with anti-inflammatory, antioxidant, and antibacterial properties [30,31]. Additionally, it contains a significant amount of fiber, working as a natural laxative [14]. The positive effects of soluble fiber on the overall symptoms of IBS-C are already known [14]. In the study included, the total consumption per day was 60g of flixweed and 90g of fig [14].

Pourmasoumi et al. investigated the symptom control of flixweed and fig in 142 patients (48 flixweed, 46 fig, 48 control) diagnosed with IBS-C using the Rome III criteria in a 4-month, single-blind, randomized, controlled trial. The study revealed a significant improvement for both the flixweed and fig groups compared to the control group regarding the IBS Severity Score System, quality of life, abdominal pain frequency, interference of life, and dissatisfaction with bowel habits. There were no significant benefits for abdominal pain severity and C-reactive protein levels. No significant difference was detected between the flixweed and fig groups; thus, no priority can be given to either of the two treatments [14].

### 3.5. A Combination of Quebracho, Conker Tree, and M. balsamea Willd Extract for IBS-C

Quebracho extract contains tannins as the active substances. They are large delocalized flavonoid structures that might have a dual function: acting as a molecular absorbent for excess hydrogen and methane and disrupting and destroying bacterial lipid layers [15]. Conker Tree extract contains escins, also known as saponins. These substances have been shown to act as antimicrobial agents and promote intestinal motility, also potentially directly reducing methane production or emission [15]. M. balsamea Willd extract contains peppermint oil which might help with abdominal discomfort [15]. The study included a blended extract of Quebracho (150 mg), Conker Tree (470 mg), and *M. balsamea Willd* oil (0.2 mL) [15].

Brown K et al. investigated the effectiveness of a blended extract of Quebracho, Conker Tree, and *M. balsamea Willd* in a 2-week, single-site, randomized, double-blind, placebo-controlled study on 16 patients (8 blended extract, 8 placebo) with IBS-C diagnosed using the Rome III criteria. At the end of the study, there was a significant improvement in constipation and bloating, with no adverse effects reported. Although there was significant improvement, the study was limited by the small number of participants [15].

### 3.6. Kiwifruit and IBS-C

Kiwifruit contains 2–3% dietary fiber and is believed to have laxative properties, significantly improving stool consistency in healthy elderly and chronically constipated adults. The included research used the daily dose of two Hayward green kiwifruits (*Actinida deliciosa*) [16].

The efficacy of Kiwifruit was evaluated in a 6-week, with restricted randomization (3:1), placebo-controlled trial on 60 patients diagnosed with IBS-C using the Rome III criteria (45 Kiwifruit 15 placebo) and 16 healthy patients as the positive control group. Out of the 70 patients who completed the study, 65 were female. There was a significant improvement in defecation frequency and colon transit time in the Kiwifruit group. There were no statistically significant modifications in fecal volume change, life stress, or post-defecation feelings [16].

### 3.7. Modified Sinisan for IBS-C

Sinisan is a traditional Chinese medicine used in a variety of gastrointestinal disorders. The study included used a modified version of Sinisan (Thorowax root 10 g, immature bitter orange 10 g, Aucklandia root 10 g, Spice-bush root 10 g, Bighead atractylodes rhizome 20 g, White peony root 10 g, *Ligusticum chuanxiong* rhizome 10 g, Chinese angelica root 10 g, and licorice 5 g water decocted) [17]. The herbal mix enhances stomach emptying and intestinal peristalsis, promoting intestinal gas discharge and regulating intestinal dilation and constriction [17]. Thorowax root contains numerous bioactive compounds such as essential oils, triterpenoid saponins, polyacetylenes, flavonoids, lignans, fatty acids, and sterols [32]. Bitter orange acts through p-Synephrine, while Aucklandia acts through costunolide, dehydrocostus lactone, dihydrocostunolide, costuslactone, α-costol, Saussurea lactone, and costuslactone [33,34,35]. The main active compounds found in atractylodes rhizome are made up of sesquiterpenes [35]. Monoterpenes, flavonoids, phenols, and tannins are found in peony root [36]. *Ligusticum chuanxiong* rhizome contains numerous phenols, organic acids, phthalides, alkaloids, polysaccharides, ceramides, and cerebrosides [37]. Chinese angelica root acts through various essential oils such as ligustilide, butylphthalide, and senkyunolide A, phthalide dimers, organic acids, and their esters such as ferulic acid, coniferyl ferulate, polyacetylenes, vitamins and amino acids [38]. The main active compounds found in licorice are flavonoids (flavanones, chalcones, isoflavanes, isoflavenes, flavones, and isoflavones), saponins (glycyrrhizic acid), phenols (liquiritin, isoliquiritin, liquiritin apioside, and isoprenoid-substituted flavonoids, chromenes, coumarins, and dihydrostilbenes), coumarins (liqcoumarin, glabrocoumarone A and B, herniarin, umbelliferone, and glycyrin) and other compounds such as fatty acids, phenol, guaiacol, asparagines, glucose, sucrose, starch, polysaccharides, and sterols (β-sitosterol, dihydrostigmasterol) [39].

An 8-week, randomized, controlled study investigated the effects of a modified Sinisan formula compared to Cisapride on 47 patients diagnosed with IBS-C using the Rome diagnostic standard (24 Sinisan, 23 Cisapride) and 22 healthy patients. Sinisan had a significant difference in efficacy, symptom scoring, and recurrence rate compared to Cisapride. Additionally, there was a statistically significant improvement in rectal threshold feeling, maximal tolerance volume, and rectum compliance in the treatment group. On the other hand, there was no difference in anal resting, systolic or diastolic pressure [17]. Nevertheless, it is worth mentioning that increased evidence regarding the subsequent adverse effects of Cisapride, mainly cardiac toxicity, has prompted the Food and Drug Administration to remove the drug from the market [40,41,42].

### 3.8. Geraniol for IBS-C

Geraniol has already been shown as beneficial in patients with IBS, by having an eubiotic effect on the gut microbiota (GM) [18]. Geraniol is a monoterpenic alcohol commonly found in essential oils, with biological effects such as antimicrobial and antitumoral [43]. Thus, Ricci et al. conducted an RCT to assess the efficacy of geraniol when it comes to symptom severity, GM composition, and inflammatory markers in patients with IBS [18]. The study concluded that patients in the geraniol group showed a significant reduction in IBS-SSS at the end of the trial, as well as an increase of the subspecies *Ruminococcaceae, Oscillospira,* and a decrease of *Erysipelotrichaceae* and *Clostridiaceae* families in the GM. Nevertheless, no statistically significant changes in the inflammatory biomarkers have been observed [18].

## 4. Discussion

Although it is one of the most prevalent functional gastrointestinal disorders, IBS still poses a great diagnostic and therapeutic challenge. Manifesting in various forms and subtypes, IBS-C particularly impacts the quality of life of patients, requiring a more personalized and tailored therapy. This has made possible the rapid development and growth in popularity of various non-pharmacological therapies and agents aimed at improving the symptoms of patients with IBS.

To this extent, more and more patients and turning towards unconventional means, such as herbal remedies, with their laxative, anti-inflammatory, and antispasmodic properties [44,45]. However, some alternative therapies such as peppermint oil and probiotics have already been suggested by some, while other herbal remedies are still not recommended [46]. The fact that IBS is such a heterogenous disease, manifesting in many ways and greatly impacting the quality of life of patients, many efforts have been made so far to find suitable therapies for the control of these symptoms [47,48]. Thus, Ried et al. developed an herbal formula that has been administered to patients with IBS over 16 weeks, the study, resulting in a significant improvement when it comes to the upper and lower gastrointestinal symptoms, such as abdominal pain, indigestion, pyrosis, constipation, diarrhea [47].

CHM has also been gaining popularity in recent years, a phenomenon accompanied by an increased research interest as well [49,50]. Thus, studies have underlined a series of herbal remedies potentially beneficial in the treatment of IBS-D, such as Jianpiwenshen therapy, ameliorating clinical symptoms and causing little to no adverse effects [50]. On the other hand, when it comes to IBS-C, several CHM formulas have been analyzed in clinical trials, with various results [12,51]. Although both studies reported an improvement when it comes to bowel habits and frequency, only one trial showed an amelioration in symptoms such as abdominal pain and distension [51]. 

Another popular non-conventional therapy for IBS consists of Western herbal medicines, with formulas with various constituents and effects on symptoms [15,52]. Of particular significance are peppermint oil and various aloe extracts, which have been proven beneficial in reducing the symptoms of IBS, particularly abdominal pain, constipation, and bloating [15,52]. Moreover, other herbal remedies and formulas have also shown promising results in small trials, although, to be able to extrapolate and implement these results, larger clinical trials are needed. Over-the-counter (OTC) medications are largely popular for the management of constipation as well [53]. Thus, certain remedies and substances such as polyethylene glycol (PEG) and senna are largely trusted and used for alleviating the symptoms of patients with constipation. Less evidence exists in the case of psyllium, magnesium salts, or fruit-based laxatives, such as kiwi, mango, or prunes [53].

Additionally, there are also several herbal remedies and formulas whose effects on IBS are currently being investigated in pilot studies [51,54]. One such example is Enterofytol ^®^, a bio-optimized extract consisting of 42 mg of curcumin and 25 mg of fennel essential oil. A 60-day observational, prospective, non-controlled, non-randomized pilot study verified the efficacy of turmeric extracts and fennel essential oil in 211 patients diagnosed with IBS using the Rome III criteria. Out of the 211 patients, 50 had IBS-C. There was a significant reduction in the severity index and improvement in the quality of life in all subtypes, including the IBS-C group, independent of age and sex. No adverse effects were reported [54]. The efficacy and safety of another herbal IBS-C formula were researched in a 3-week, two-arm, open-label, uncontrolled pilot study on 8 female patients diagnosed with IBS-C using the Rome II criteria. It consisted of lactulose 6 g, slippery elm 14 g, licorice 3 g, and oat bran 4 g daily. The treatment showed a significant improvement in daily bowel movements, stool consistency, sense of straining, abdominal pain severity, bloating severity, and global symptom severity. There was no significant difference in the sense of urgency and flatulence severity. Additionally, there were no reported side effects, significant changes in vital signs, or altered laboratory results. Although the improvements were significant, the study was limited by the small number of participants who were all females, and the lack of a control group [51].

Due to its significant impact on patients’ quality of life and the limited success of many pharmaceutical treatments in producing desired outcomes, numerous individuals suffering from IBS have started searching for alternative remedies. The desire for a holistic strategy encompassing non-pharmaceutical remedies for IBS stems from patients’ yearning for thorough and individualized healthcare that takes into account the various factors influencing their well-being. As a result of the ongoing introduction of diverse non-pharmacological solutions, healthcare experts should acknowledge the potential positive impact of these remedies, rather than ruling them out solely due to insufficient or limited evidence. Physicians need to acknowledge IBS’s multifactorial nature, honor patient preferences, and cooperate with them to devise comprehensive treatment strategies that emphasize overall wellness and the management of symptoms.

Nevertheless, a significant number of concerns regarding herbal remedies persist and are yet to be addressed, such as the lack of regulations, standardized manufacturing processes, and studies regarding pharmacokinetics and pharmacodynamics. Thus, more research is needed to be able to assess the beneficial effects of these compounds and eventually implement them into clinical practice.

When conducting this systematic review, it became evident that the available literature concerning the efficacy of herbal remedies in the context of IBS-C is notably limited. Although we identified and analyzed four existing systematic reviews exploring the broader landscape of herbal interventions for IBS, a notable paucity of systematic reviews focusing exclusively on IBS-C was observed. This observation underscores the need for further comprehensive investigations tailored specifically to this subtype of IBS, as the clinical characteristics and underlying pathophysiology of IBS-C may differ from other subtypes. The limited availability of systematic reviews tailored to IBS-C highlights a critical research gap, therefore warranting future research endeavors that rigorously assess the efficacy, safety, and potential mechanisms of action of herbal remedies in addressing the unique challenges posed by this distinct IBS subtype. Such dedicated investigations hold the potential to provide valuable insights and evidence-based guidance for both clinicians and patients navigating the management of IBS-C, which remains a clinically relevant and often challenging condition to address. Furthermore, an important consideration arising from this systematic review is the relatively limited number of patients included in most of the available studies investigating the efficacy of herbal remedies in the context of IBS-C. Most studies identified in our review exhibited small sample sizes, which can introduce variability and limit the generalizability of their findings. This underscores the necessity for well-designed, adequately powered clinical trials specifically tailored to address the nuances of IBS-C. Large-scale clinical trials offer several advantages, including improved statistical power to detect meaningful treatment effects, better characterization of safety profiles, and the ability to stratify patient subgroups for more personalized insights. Such trials are essential to provide robust and reliable evidence that can guide clinical practice and enhance our understanding of the potential benefits and risks associated with herbal remedies in the management of IBS-C. Future research endeavors should prioritize the execution of rigorous clinical trials to address these limitations and strengthen the evidence base, ultimately benefiting both healthcare providers and individuals grappling with the challenges of IBS-C management.

## 5. Conclusions

This systematic review suggests that herbal remedies may have potential benefits in the management of IBS-C. Notwithstanding, the constraints inherent in the caliber of available limited evidence necessitate the undertaking of additional randomized controlled trials (RCTs) to substantiate the effectiveness and safety of distinct herbal interventions tailored to IBS-C. Clinicians should exercise caution when recommending herbal remedies and consider individual patient characteristics and preferences.

## Figures and Tables

**Figure 1 nutrients-15-04216-f001:**
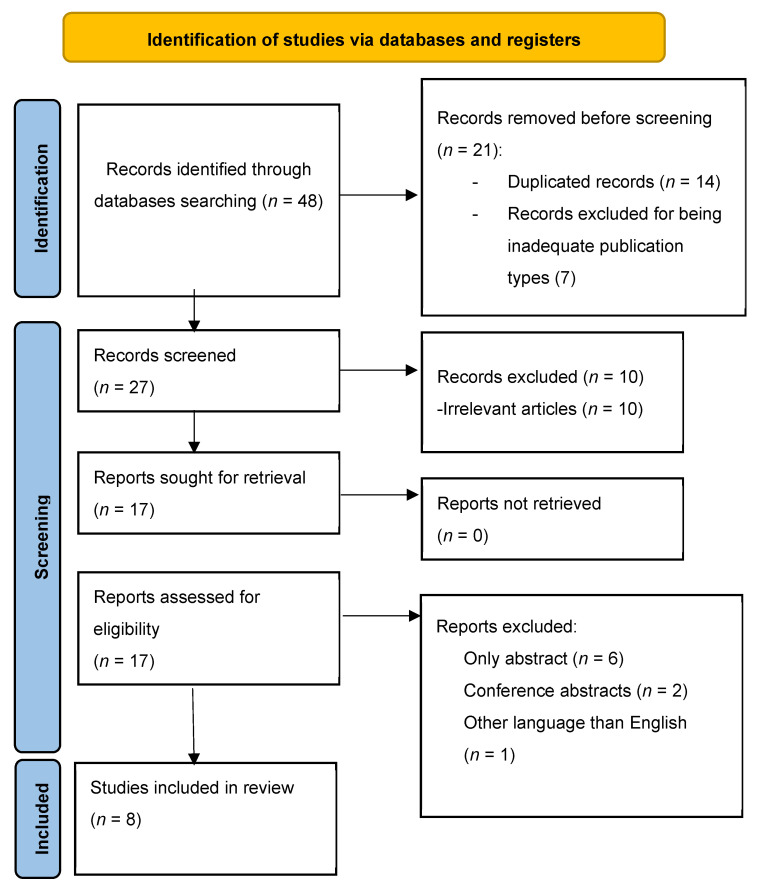
PRISMA flowchart of the included studies.

**Table 1 nutrients-15-04216-t001:** Studies assessing various herbal remedies in IBS-C.

Author	Study Design	Participants	Intervention	Main Findings
Sallon et al. (2002) [11]	Randomized, double-blind, placebo-controlled clinical trial	61 patients with IBS-C	Padma Lax vs. placebo for 12 weeks 2 tablets/day or 1 tablet/day if diarrhea or loose stools occurred. 1 tablet (482 mg) contained: Aloes standardized extract (12 mg) Calumbae root (10 mg) Condurango bark (10 mg) Frangula bark (52.5 mg) Gentian root (35 mg) Elecampane rhizome (35 mg) Tropical almond fruit (35 mg) Long pepper (3.5 mg) Cascara bark (52.5 mg) Chinese rhubarb root (70 mg) Nux vomica seed (1.75 mg) Ginger root (70 mg). It also contained 3 non-herbal elements: Heavy kaolin (25 mg) Sodium bicarbonate (15 mg) Sodium sulfate (35 mg). The excipients used were: Silicium dioxide (2.4 mg) Magnesium stearate (4.6 mg).	Statistically significant improvements were found in: Constipation severity Number of daily bowel movements Abdominal pain affecting daily activities Presence of moderate or severe pain Abdominal distention severity Incomplete evacuation Flatulence. There were no significant improvements in: Abdominal distention prevalence Abdominal pain severity. 10 patients complained of mild side effects: Slight headache (1 patient) Nausea (1 patient) Hoarseness (1 patient) Diarrhea or loose stools (7 patients) with rapid resolution when lowering dose from 2 tablets/day to 1 tablet/day.
Bensoussan et al. (2015) [12]	Two-arm, randomized, double-blind, placebo-controlled clinical trial	125 patients with IBS-C	CHM vs. placebo for 8 weeks 10 capsules (4.2 g)/day, 5 in the morning and 5 in the night containing: *Paeonia lactiflora PALL.*, *radix* (23%) *Citrus aurantium* L., *Fructus immaturas* (20%) *Magnolia officinalis REHD.* and *WILS*., *cortex* (14.5%) *Citrus reticulata* L., *pericarpi-um* (14.5%) *Glycyrrhiza uralensis FISCH.*, root (11%) *Rheum palmatum L.*, *radix* (10%) *Atractylodes lancea* (*THUNB.*) *DC*, rhizome (7%).	There were statistically significant changes at 8 weeks in: Adequate relief (PP) Bowel habits Frequency of hard lumpy stools Bothersomeness of hard lumpy stools Straining during bowel movement Stool form Physical functioning. There were statistically significant changes at 16 weeks in: Physical functioning. There was no significant improvement at 8 weeks in: Adequate relief (ITT). There was no significant improvement at 16 weeks in: Adequate relief (PP, ITT) Bowel habits Frequency of hard lumpy stools Bothersomeness of hard lumpy stools Straining during bowel movement Stool form Abdominal pain Abdominal distension Interference with life Overall quality of life Depression Anxiety Stress Work productivity. 6 patients complained of moderate side effects: Diarrhea (3 patients) Blood in urine (1 patient) Increased bloating (1 patient) Dizziness and headaches (1 patient).
Pazhouh et al. (2020) [13]	Block-randomized, double-blind, placebo-controlled clinical trial	70 patients with IBS-C	PHS vs. placebo for 6 weeks 5 mL every 8 h of the syrup prepared with: *Artemisia Absinthium* (100 mg) *Cuscuta Campestris* (100 mg) *Cassia Fistula* (230 mg) *Echium Amoenium* (270 mg) *Mellisa Officinalis* (190 mg) British Pharmacopoeia syrup to make the mixture 100 mL.	There was a significant improvement in: Positive responses to treatment Severity score Bristol Stool Score. There was no significant improvement in: Anxiety score Depression score. 4 patients complained of negligible side effects: Headache (2 patients) Drowsiness (1 patient) Increase in menstrual bleeding (1 patient).
Pourmasoumi et al. (2019) [14]	Single-blind, randomized, placebo-controlled clinical trial	142 patients with IBS-C	Flixweed vs. fig vs. placebo for 4 months 60 g flixweed and 90 g fig/day, half before breakfast and half before lunch.	Statistically significant improvements were found in: IBS Severity Score System Quality of life Abdominal pain frequency Interference of life Dissatisfaction with bowel habits. No significant difference was found in: Abdominal pain severity C-reactive protein levels. There was no statistically significant difference between flixweed and fig. No adverse effects were reported.
Brown et al. (2015) [15]	Single-site, randomized, double-blind, placebo-controlled clinical trial	16 patients with IBS-C	Blended extract of Quebracho, Conker Tree, and *M. balsamea Willd* vs. placebo for 2 weeks Extract consisting of: Quebracho (150 mg) Conker Tree (470 mg) *M. balsamea Willd* oil (0.2 mL).	There was a significant improvement in: Constipation Bloating. No side effects were reported.
Chang et al. (2010) [16]	Restricted randomization (3:1), placebo-controlled clinical trial	76 patients, 60 with IBS-C and 16 healthy	Kiwifruit vs. placebo for 4 weeks 2 Hayward green kiwifruits/day.	Statistically significant improvements were found in: Defecation frequency Colon transit time. No significant improvement was found in: Fecal volume change Life stress Post-defecation feelings. No adverse effects were reported.
Yu et al. (2005) [17]	Randomized, placebo-controlled clinical trial	47 patients with IBS-C	Modified Sinisan formula vs. Cisapride for 8 weeks 1 dose divided into 2 portions consisting of: Thorowax root (10 g) Immature bitter orange (10 g) Aucklandia root (10 g) Spice-bush root (10 g) Bighead atractylodes rhizome (20 g) White peony root (10 g) *Ligusticum chuanxiong* rhizome (10 g) Chinese angelica root (10 g) Licorice (5 g) water decocted.	Statistically significant differences were found in: Efficacy Symptom scoring Recurrence rate Rectal threshold feeling Maximal tolerance volume Rectum compliance. No significant difference was found in: Anal resting pressure Anal systolic pressure Anal diastolic pressure. No side effects are mentioned.
Ricci et al. (2022) [18]	interventional, prospective, multicentric, randomized, double-blinded, placebo-controlled trial	56 patients with IBS	Geraniol vs. placebo for 4 weeks 1 capsule (470 mg)/day consisting of: Palmrose EO high geraniol (90 mg) Pulverized Zyngiber officinalis root (360 mg). The excipients used were: Vegetal magnesium stearate (10 mg) Silicon dioxide (10 mg). IBS-SSS GM composition Inflammatory markers	There was a statistically significant difference in: Abdominal Pain Days with abdominal Pain in the last 10 Bloating Satisfaction bowel habits IBS-SSS Score IBSS-SSS Score variations Responders (reduction 50 points IBS-SSS). There was no difference in: Interference with daily activities. 2 patients complained of side effects, 1 of which were unspecified gastric symptoms. Significant reduction in IBS-SSS in the geraniol group. Increase in the Ruminococcaceae and Oscillospira species and decrease in the Erysipelotrichaceae and Clostridiaceae families. No changes when it comes to inflammatory biomarkers.

CHM: Chinese Herbal Medicine; IBS-SSS: IBS Symptom Severity Scale; PHS: Persian herbal syrup; GM: Gut Microbiota.

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
