# Peer review of "Herbal Remedies for Constipation-Predominant Irritable Bowel Syndrome: A Systematic Review of Randomized Controlled Trials"

_nutrients, 2023, doi:10.3390/nu15194216_

Round 1

Reviewer 1 Report

Dear Authors

I appreciate your work Constipation is a frequent problem in the clinic, particularly in the context of IBS.
The use of herbal medication is an increasingly frequent trend, and it is often difficult to assess its effectiveness.
I consider the work to be relevant and I just made a few small notes about the use of commas. A comment will be necessary regarding cisapride. And, perhaps, a comment on the availability of some (oriental) products in the west.

Many regards

Author Response

Dear Reviewer,

Thank you very much for your comments and observations regarding our manuscript. We have made the necessary changes in accordance to your comments.

Best regards,

The authors

Reviewer 2 Report

Th author has tried to present a systematic review of all the herbal medicines and remedies that have evidential findings of their benefits to mainly improve IBS-C outcomes. However, the paper has following shortcomings:

1. It seems like 11 research articles are not enough to present a conclusive finding in this review. The authors are adviced to add more recent studies (after 2015). Many studies or trials included too few patients/subjects , that brings to my attention that author has not attempted to make any exclusion based on study participants. The authors are advised to only include those studies with more than 50 patients/participants.

2. The authors are not clear in their inclusion and exclusion criteria. Please write separate paragraphs on both of these criteria, and explain clearly why certain records were excluded. What kind of automated tools were used to make exclusions?

3. Figure 1 has to be remake. It should be organized, diligently presented. Please refer other studies to properly present a PRISMA flowchart.

4. Who are those reviewers who did study selection, data extraction and other pieces in this study. Please mention their names, and what is their relation with this manuscript. Are these reviewers endorsing any products or working in the company who sells or endorsed any of these herbal products?

5. Table 1 has to be expanded with more studies/clinical trial and research papers. Add study design type, formulation and dosage details of each herbal product. Findings should be more specific. Wordings like almost all parameters are used are very casual. Please be specific. Please include those studies which have done IBS-QOL and IBS-SSS assessment along with clinical outcomes of IBS cases. Please be very specific about side-effects.

6. Paragraphs about all the herbal remedies should be re-written. Focus should be on the bioactive components found in these herbal remedies, and what study have been done using only bio-active components. For example, Actinidin is a bio-active component found in kiwi-fruit, and curcumin found in turmeric. Please try to mention important studies which have used bio-active components only to treat/improve IBS-C cases. Similarly, what type of specific lipids or flavonoids or glycosides have been found in the flixweed, and other herbal remedies. Please be specific about these components, enzymes, bio-active molecules, and their uses and benefits and research involving them. Please make this section very informative and try to include more recent articles in the references. 

These changes may not be final, and further changes can be asked to revise this manuscript. 

English language is fine, but there are typos and spelling mistakes on some places. 

Author Response

Response to Reviewer 2 Comments

Point 1: It seems like 11 research articles are not enough to present a conclusive finding in this review. The authors are adviced to add more recent studies (after 2015). Many studies or trials included too few patients/subjects , that brings to my attention that author has not attempted to make any exclusion based on study participants. The authors are advised to only include those studies with more than 50 patients/participants.

Response 1: We found 4 systematic reviews about the efficiency of herbal remedies in IBS. Nevertheless, there are no systematic reviews about the efficiency of herbal remedies only in constipation predominant IBS. For this reason, the number of studies included in our review is limited. We added a paragraph in the discussion chapter, in which we explain this aspect.

Point 2: The authors are not clear in their inclusion and exclusion criteria. Please write separate paragraphs on both of these criteria, and explain clearly why certain records were excluded. What kind of automated tools were used to make exclusions?

Response 2: The inclusion and exclusion criteria were in accordance with the Preferred Reporting Items for Systematic Reviews and Meta-Analyses (PRISMA) guidelines to ensure transparency and completeness in reporting the review process. We did not use automation tools. We did not use automation tools for the exclusion criteria. It was a mistake in the PRISMA flowchart. We corrected our mistake. Thank you!

Point 3: Figure 1 has to be remake. It should be organized, diligently presented. Please refer other studies to properly present a PRISMA flowchart.

Response 3: We modified the PRISMA flowchart, and we corrected the errors.

Point 4: Who are those reviewers who did study selection, data extraction and other pieces in this study. Please mention their names, and what is their relation with this manuscript. Are these reviewers endorsing any products or working in the company who sells or endorsed any of these herbal products?

Response 4: Thank you for your observation. We are sorry for the confusion, the study selection, data extraction, and quality assessment of the included studies have been performed independently by two of the authors of our study (Stefan Lucian Popa and Abdulrahman Ismaiel). We have added the abbreviations in brackets in the Materials and Methods section of our manuscript. Moreover, we report no conflict of interests, because none of the authors are affiliated to any pharmaceutical companies, thus not endorsing any products or herbal remedies included in our systematic review.

Point 5: Table 1 has to be expanded with more studies/clinical trial and research papers. Add study design type, formulation and dosage details of each herbal product. Findings should be more specific. Wordings like almost all parameters are used are very casual. Please be specific. Please include those studies which have done IBS-QOL and IBS-SSS assessment along with clinical outcomes of IBS cases. Please be very specific about side-effects.

Response 5: Thank you for your thorough observations. We have updated our table accordingly.

Point 6: Paragraphs about all the herbal remedies should be re-written. Focus should be on the bioactive components found in these herbal remedies, and what study have been done using only bio-active components. For example, Actinidin is a bio-active component found in kiwi-fruit, and curcumin found in turmeric. Please try to mention important studies which have used bio-active components only to treat/improve IBS-C cases. Similarly, what type of specific lipids or flavonoids or glycosides have been found in the flixweed, and other herbal remedies. Please be specific about these components, enzymes, bio-active molecules, and their uses and benefits and research involving them. Please make this section very informative and try to include more recent articles in the references. 

Response 6: Thank you for your comment. We have modified the sections about the composition, bioactive compounds and their effects on every included substance.

Reviewer 3 Report

After careful reviewing of the submitted manuscript, my attention was drawn to following aspects:

1. The first noticeable problem is the temporal range of the bibliographic sources. Of the 11 publications selected for review, 7 are older than 5 years. It is important to rely on the latest research data in any rapidly developing scientific field, including medicine. In its current form, the manuscript does not add significant novelty to the discussion on the use of herbal preparations in irritable bowel syndrome with predominant constipation.

2. Only PubMed was included in the abstract and PRISMA protocol, while other databases are mentioned in line 154. Limiting oneself to one database may affect the incompleteness and one-sidedness of the review. Contemporary research often requires multifaceted analysis, using multiple databases to ensure the reliability of the review.

3. The literature should  be more up-to-date, also in the Introduction section. Articles selected by the authors for the review should be fully coincided with the title of the manuscript. E.g. publication 20 from 2011 talks about constipation, but not in the context of IBS.

4. In line 238, I noticed a typographical error - probably the number '11' should appear instead of the number '1'.

5. It is  needed to use italics for Latin names. This is standard in scientific literature and should be used consistently throughout the text.

6. In line 391, the word "geraniol" is capitalized rather than lowercase.

I appreciate the work the authors put into writing the manuscript, but I don't think it's ready to publish in its present form. I encourage the authors to revise and update their bibliographic sources to bring the work into current requirements for scientific publications.

Author Response

Response to Reviewer 3 Comments

Point 1: The first noticeable problem is the temporal range of the bibliographic sources. Of the 11 publications selected for review, 7 are older than 5 years. It is important to rely on the latest research data in any rapidly developing scientific field, including medicine. In its current form, the manuscript does not add significant novelty to the discussion on the use of herbal preparations in irritable bowel syndrome with predominant constipation.

Response 1: We also explained that clinical trials are necessary because of the small number of patients included in most available studies. We added a paragraph in the discussion chapter where we explain this aspect.

Point 2: Only PubMed was included in the abstract and PRISMA protocol, while other databases are mentioned in line 154. Limiting oneself to one database may affect the incompleteness and one-sidedness of the review. Contemporary research often requires multifaceted analysis, using multiple databases to ensure the reliability of the review.

Response 2: Thank you for your observation. We have made a mistake regarding the elaboration of the PRISMA flowchart. All the named databases were searched, and the number in the flowchart represents the number of potential articles across all named databases, not only Pubmed.

Point 3: The literature should  be more up-to-date, also in the Introduction section. Articles selected by the authors for the review should be fully coincided with the title of the manuscript. E.g. publication 20 from 2011 talks about constipation, but not in the context of IBS.

Response 3: We are sorry for the error. We have corrected the mistake accordingly.

Point 4: In line 238, I noticed a typographical error - probably the number '11' should appear instead of the number '1'.

Response 4: Thank you for your observation. We have corrected the mistake.

Point 5: It is  needed to use italics for Latin names. This is standard in scientific literature and should be used consistently throughout the text.

Response 5: Thank you for your observation. We have updated the formatting of Latin names throughout the text.

Point 6: In line 391, the word "geraniol" is capitalized rather than lowercase.

Response 6: Thank you for your observation, we have corrected the mistake.

Reviewer 4 Report

I’ve read with attention the systematic review by Chiarioni et al. that is potentially of interest. The background and aim of the study have been clearly defined. The methodology applied is overall correct, the results are reliable and adequately discussed. I’ve only some minor comments:

- It is not clear why the authors excluded non dietary fibres from the systematic review. For instance, psyllium and guar are herbal products.

- Table 1: phyllum should be psyllium

Author Response

Response to Reviewer 4 Comments

Point 1: It is not clear why the authors excluded non dietary fibres from the systematic review. For instance, psyllium and guar are herbal products.

Response 1: We could not find enough evidence about psyllium or other herbal remedies. We mention this in line 433: Less evidence exists in the case of psyllium, magnesium salts, or fruit-based laxatives, such as kiwi, mango, or prunes. No evidence for guar was found.

Point 2: Table 1: phyllum should be psyllium

Response 2: Thank you for your observation. We have corrected the mistake.

Round 2

Reviewer 3 Report

Upon a meticulous reviewing of the submitted new revision of the manuscript, I wish to provide the following feedback:

1.      I recommend reconsidering the current title of the manuscript. Specifically, it would be beneficial if the title more explicitly reflected the emphasis on clinical trials. This would ensure greater clarity and precision for readers who are seeking content specifically related to this domain.

2.      I observed that the authors have retained a mention of only a single database within the abstract. It may give an incomplete representation of the breadth of sources consulted in this research. Additionally, the PRISMA flowchart provided in the methodology section still suggests the use of a singular database.

In accordance of the above suggestions, I believe that addressing these points will further improve the quality and clarity of the manuscript. By ensuring that the title accurately reflects the content and providing full transparency of data sources, the manuscript can make a solid contribution to the academic discourse in its field.

Author Response

Response to Reviewer 3 Comments

Point 1: I recommend reconsidering the current title of the manuscript. Specifically, it would be beneficial if the title more explicitly reflected the emphasis on clinical trials. This would ensure greater clarity and precision for readers who are seeking content specifically related to this domain.

Response 1: Thank you for your observation. Upon carefully reviewing our manuscript, we have decided to only include the randomized controlled trials intro our systematic review (8 studies). Thus, the other two studies, being pilot studies, have been moved in the Discussions chapter of our manuscript. Therefore, we have modified the title of our manuscript in order to accurately depict the topic of our review.

Point 2: I observed that the authors have retained a mention of only a single database within the abstract. It may give an incomplete representation of the breadth of sources consulted in this research. Additionally, the PRISMA flowchart provided in the methodology section still suggests the use of a singular database.

Response 2: Thank you for your observation. We have modified the Abstract accordingly. We are sorry for the mistake. As previously mentioned, we have searched PubMed, MEDLINE, Embase, Scopus, and Cochrane Library. We have also modified the PRISMA flowchart accordingly.